# Self-Assembling Injectable Hydrogel for Controlled Drug Delivery of Antimuscular Atrophy Drug Tilorone

**DOI:** 10.3390/pharmaceutics14122723

**Published:** 2022-12-06

**Authors:** Mohamed M. Abdelghafour, Ágota Deák, Tamás Kiss, Mária Budai-Szűcs, Gábor Katona, Rita Ambrus, Bálint Lőrinczi, Anikó Keller-Pintér, István Szatmári, Diána Szabó, László Rovó, László Janovák

**Affiliations:** 1Department of Physical Chemistry and Materials Science, University of Szeged, Rerrich Béla tér 1, H-6720 Szeged, Hungary; 2Department of Chemistry, Faculty of Science, Zagazig University, Zagazig 44519, Egypt; 3Institute of Pharmaceutical Technology and Regulatory Affairs, University of Szeged, Eötvös str. 6., H-6720 Szeged, Hungary; 4Institute of Pharmaceutical Chemistry, University of Szeged, Eötvös str. 6, H-6720 Szeged, Hungary; 5Department of Biochemistry, Albert Szent-Györgyi Medical School, University of Szeged, H-6720 Szeged, Hungary; 6Department of Oto-Rhino-Laryngology and Head & Neck Surgery, University of Szeged, Tisza Lajos krt. 111, H-6724 Szeged, Hungary

**Keywords:** thiolated chitosan, modified PVA, tilorone, injectable self-assembled hydrogel, mucoadhesive hydrogel, prolonging drug release

## Abstract

A two-component injectable hydrogel was suitably prepared for the encapsulation and prolonged release of tilorone which is an antimuscular atrophy drug. The rapid (7–45 s, depending on the polymer concentration) in situ solidifications of the hydrogel were evoked by the evolving Schiff-base bonds between the aldehyde groups of modified PVA (4-formyl benzoate PVA, PVA-CHO, 5.9 mol% functionalization degree) and the amino groups of 3-mercaptopropionate chitosan (CHIT-SH). The successful modification of the initial polymers was confirmed by both FTIR and NMR measurements; moreover, a new peak appeared in the FTIR spectrum of the 10% *w*/*v* PVA-CHO/CHIT-SH hydrogel at 1647 cm^−1^, indicating the formation of a Schiff base (–CH=N–) and confirming the interaction between the NH_2_ groups of CHIT–SH and the CHO groups of PVA-CHO for the formation of the dynamic hydrogel. The reaction between the NH_2_ and CHO groups of the modified biopolymers resulted in a significant increase in the hydrogel’s viscosity which was more than one thousand times greater (9800 mPa·s) than that of the used polymer solutions, which have a viscosity of only 4.6 and 5.8 mPa·s, respectively. Furthermore, the initial chitosan was modified with mercaptopropionic acid (thiol content = 201.85 ± 12 µmol/g) to increase the mucoadhesive properties of the hydrogel. The thiolated chitosan showed a significant increase (~600 mN/mm) in adhesion to the pig intestinal membrane compared to the initial one (~300 mN/mm). The in vitro release of tilorone from the hydrogel was controlled with the crosslinking density/concentration of the hydrogel; the 10% *w*/*v* PVA-CHO/CHIT-SH hydrogel had the slowest releasing (21.7 h^−1/2^) rate, while the 2% *w*/*v* PVA-CHO/CHIT-SH hydrogel had the fastest releasing rate (34.6 h^−1/2^). Due to the characteristics of these hydrogels, their future uses include tissue regeneration scaffolds, wound dressings for skin injuries, and injectable or in situ forming drug delivery systems. Eventually, we hope that the developed hydrogel will be useful in the local treatment of muscle atrophy, such as laryngotracheal atrophy.

## 1. Introduction

Injection-based, minimally invasive hydrogel delivery has recently garnered a lot of attention in the biomedical field. Injectable hydrogels are often generated through a rapid sol–gel phase transition or an in situ chemical crosslinking. They can be injected directly into the desired sites [1,2,3]. Based on the gelling mechanism, injectable hydrogels are classified as chemically and physically crosslinked hydrogels [1,2,3]. Chemically crosslinked injectable hydrogels are created by forming covalent bonds between polymer chains via Diels–Alder reactions, Michael additions, enzyme mediations, Schiff-base reactions, or photopolymerizations [1,2,3]. Physically crosslinked injectable hydrogels, on the other hand, are generated via intermolecular interactions, such as hydrogen bonds, van der Waals forces, hydrophobic interactions, π interactions, ionic bonds, and host–guest interactions [1,2,3]. Hydrogels that have been physically crosslinked typically offer a friendly environment for cells and bioactive compounds. They typically exhibit a very low mechanical strength, and external stimuli can rapidly change their morphology and characteristics. Contrarily, chemically crosslinked hydrogels have demonstrated a comparatively high mechanical strength and stability [2,3]. In comparison to their preformed hydrogels, injectable hydrogels offer many advantages, including easy delivery by syringe and minimal surgical wounds, convenient synthesis, versatility, a high drug-loading capacity, and a controlled drug-release ability [3]. Injectable hydrogels can be loaded with a diverse range of substances, including cytokines, immune cells, antibodies, vaccines, and chemotherapeutic and immunotherapeutic agents [2,3,4,5]. It will be possible to design and improve novel therapeutic injectable hydrogels by successfully combining components from various systems.

Recent developments in biomedical applications have focused a lot of interest on hydrogels based on their dynamic linkages [6,7,8,9]. Aldehyde and amine groups combine to create Schiff-base (imine) bonds (-N=CH-). A dynamic equilibrium can exist between the formed Schiff base and the aldehyde and amine reactants in aqueous solutions because aromatic Schiff bases are more stable than their aliphatic counterparts [6]. Thus, unlike their aromatic counterparts, unstable aliphatic Schiff bases are unable to reach a dynamic equilibrium. However, the pH dependence of the dynamic equilibrium of aromatic Schiff bases causes them to move toward imine bond formation under physiological conditions (pH 7.4) [6]. Due to the dynamic equilibrium between the formed Schiff base linkage and the aldehyde and amine reactants, Schiff-base hydrogels have injectability and self-healing capabilities in their network structure as well as responsivity to several biochemical stimuli, including pH, amino acids, enzymes, and vitamin B6 derivatives, which can alter the balance of Schiff bases [6]. In contrast to conventional injectable hydrogels, self-healing hydrogels can be injected without fragmentation and can be incorporated as bulk gels at the target region [9]. As a result, in this report, benzaldehyde groups were chosen to modify the PVA chain, which subsequently reacted with the amine groups on the chitosan backbone to form the hydrogel networks without using reducing agents.

Chitosan, a polycationic biopolymer, is usually obtained from the alkaline deacetylation of chitin, the main component of the exoskeletons of arthropods and of the cell walls of fungi. Compared with ~80 million tons of the annual production of synthetic polyethylene [10], there is over 10 gigatons of chitin that is constantly present in the biosphere. Therefore, chitosan is one of the most important green and renewable materials [11]. It is generally nontoxic, biocompatible, and biodegradable and has been used for many pharmaceutical and medical purposes, such as periodontal and implant surgeries [12,13], topical ocular applications [14], and injections [15]. Some chitosan-digesting enzymes (e.g., pepsin and lipase) are common elements in physiological systems; therefore, dynamic chitosan-based hydrogels have the potential to be used as controlled drug-release vehicles. However, numerous drawbacks, such as dissolution in extremely acidic conditions, a high cost, a low surface area, and poor mechanical and thermal characteristics, limit chitosan’s use. To improve its water solubility and mucoadhesive properties, chitosan was modified with 3-mercaptopropionic acid. Polyvinyl alcohol (PVA), a water-soluble synthetic polymer, has been widely used in the creation of hydrogels due to good features such as solvent resistance, high hydrophilicity, mechanical performance, and biocompatibility [16]. It is an inert, nontoxic, biocompatible synthetic polymer that has received FDA approval for its clinical use in humans and is already used in vivo for contact lenses, eye drops, and implantable devices including catheters, occlusive agents, tissue adhesion barriers, cartilage replacements, and nerve routers [17]. Additionally, it is frequently utilized alone or in copolymers for biomedical applications [18,19]. An in vitro test revealed that PVA is biocompatible with various cell lines [20]; however, the biodegradability and/or elimination of PVA in animals is still being debated despite the optimistic results of Yamaoka et al. regarding the molecular weight dependence of the body distribution of PVA which reported that the amount of PVA accumulated in the organs was too small to affect their biological fate and only weakly interacted with different cells [21]. The urinary excretion of PVA (with a molecular weight of 70 kD) after intravenous (IV) injection was 43% within 240 min followed by gradual excretion thereafter [21]. These results indicate that PVA is excreted to the same extent as polyethylene glycol (PEG), therefore supporting the safety of PVA [21]. Additionally, Kaneo et al. studied the intravenous (IV) injection of PVA and suggested that PVAs are eliminated exclusively by mechanisms that do not involve saturable transport processes [22]. Cerroni et al. studied the in vivo biological fate and elimination pathways of poly(vinyl alcohol) microbubbles (PVA MBs) in mice, and the study confirmed the absence of cytotoxic effects. Additionally, the hypothesized mechanism of PVA MB bioelimination was attributed to macrophages, which are known to generate reactive oxygen species, which could contribute to the oxidation and degradation of PVA polymers and crosslinking networks [23]. In addition, PVA can be easily converted to benzaldehyde derivatives through the esterification reaction between 4-formyl benzoic acid and the pendant OH groups of the PVA backbone [24]. PVA with terminal CHO groups provides a good crosslink with the NH_2_ of modified chitosan to form a hydrogel and to make up for the benefits and drawbacks of both PVA and chitosan.

The focal atrophy of skeletal muscles occurs following denervation and, in the case of several muscle dystrophies, creates therapeutic challenges. Acquired laryngotracheal muscle atrophy is a potentially life-threatening condition as well as a very demanding and difficult problem in laryngology. In cases of vocal fold palsy (especially bilateral) caused by muscular atrophy, which is established through recurrent nerve injury, the most important symptom besides hoarseness is dyspnoea, which can lead to severe suffocation as well [25,26]. Several therapeutic strategies involving laryngeal framework surgery (medialization thyroplasty) and injectable laryngoplasty (lipoaugmentation, steroid, bFGF, and stem cell therapy) have been explored, but their effectiveness has been limited [27,28,29]. Notably, augmenting muscle size has therapeutic importance for laryngotracheal muscle atrophy without the correction of innervation. The BMP (bone morphogenic protein)-Smad1/5/8 signaling axis is an important positive regulator of adult myofibers and is dominant over myostatin/TGF-beta signaling [30], and increased BMP7 expression was reported to induce muscle hypertrophy [31]. Interestingly, tilorone, a synthetic, low-molecular-weight compound that can be administered orally or through an intraperitoneal injection, induces BMP (bone morphogenetic protein) signaling and enhances BMP-7 expression [32]. Tilorone was also demonstrated to have antifibrotic actions in a mouse model of silica-induced lung fibrosis [32], representing a potential novel therapy for the treatment of cardiac fibrosis associated with heart failure [33], and it also remains effective against several viruses including SARS-CoV-2 [34]. Importantly, an injection of tilorone significantly enhances muscle mass in patients with cancer cachexia [35,36]. Tilorone encapsulation in injectable self-healing hydrogels for controlled release provides several advantages, including a reduced dose, fewer side effects, increased bioavailability, a high carrier capacity, a high stability, and improved therapeutic performance. Several ongoing clinical studies are investigating the safety and efficiency of the systemic administration of myostatin inhibitors in sarcopenia, muscle dystrophies, and muscle wasting following hip fracture surgery, but the local administration of these compounds in the case of local muscle atrophy has not been studied yet.

The goal of our research was to develop an injectable mucoadhesive hydrogel incorporating tilorone, a BMP inducer, for local injection in the therapy of muscular atrophy as a potential therapeutic approach. The dynamic injectable hydrogel was physiochemically evaluated using FTIR, viscosity, mucoadhesive, self-healing, and EDX measurements. The in vitro drug-release measurements were carried out under physiological conditions (37 °C, pH 7.4), and the drug release was controlled with the hydrogel concentration/crosslinking density.

## 2. Materials and Methods

### 2.1. Materials

Polyvinyl alcohol (PVA, Mw = 46.8 kDa, 86–89% hydrolyzed) was purchased from Nagart Kft., Győrújbarát, Hungary. Low-molecular-weight (50–190 kDa) chitosan (CHIT, C_12_H_24_N_2_O_9_, ≥75% deacetylation) was purchased from Merck Ltd. (Darmstadt, Germany). The polydispersity of the PVA and chitosan was shown in Appendix A. 1-(3-Dimethylaminopropyl)-3-ethyl carbodiimide hydrochloride (EDC, C_8_H_17_N_3_·HCl, Mw 191.70, 98+%) was obtained from Thermo Fisher GmbH (Kandel, Germany), while 3-mercaptopropionic acid was purchased from Fluka Analytical (Munich, Germany). Moreover, 4-formylbenzoic acid (FBA, C_8_H_6_O_3_, 97%), N, N’-dicyclohexylcarbodiimide (DCC, C_13_H_22_N_2_, 99%), and 4-(dimethylamino)pyridine (DMAP, C_7_H_10_N_2_, ≥99%) were acquired from Sigma–Aldrich Chemie GmbH, Steinheim, Germany. Phosphate-buffered saline (PBS; pH~7.4) solution was prepared using sodium dihydrogen phosphate monohydrate (NaH_2_PO_4_·H_2_O) from Sigma–Aldrich as well as di-sodium hydrogen phosphate dodecahydrate (Na_2_HPO_4_·12H_2_O) and sodium chloride (NaCl) acquired from Molar Chemicals Kft., Hungary. Tilorone dihydrochloride (C_25_H_34_N_2_O_3_·2HCl) as an antimusclular atrophy drug was acquired from Merck KGaA, Darmstadt, Germany. All chemicals were used exactly as supplied, with no additional purification.

### 2.2. Synthesis of 4-Formylbenzoate PVA (PVA-CHO)

The PVA with CHO pendant groups was prepared using Steglich esterification method [37], which was carried out as follows: 1 g of PVA was dissolved in 15 mL of DMSO, and then 0.6 g of 4-formylbenzoic acid and 0.45 g of DMAP were added to the PVA solution, which was then stirred for 5 min before adding 0.62 g of DCC. The reaction was vigorously magnetically stirred for 24 h at room temperature. After the reaction time, the reaction mixture was filtrated through a 0.1 μm syringe filter (Whatman) to remove the insoluble byproduct of the coupling agent (DCU, N, N-dicyclohexyl urea), and then the filtrate was added to an excess amount of ethanol (~150 mL) to precipitate the crude product which was collected through centrifugation (5000 rpm × 15 min), followed by three ethanol washes, and was finally dried under vacuum. The functionalization degree of PVA with CHO was estimated through simple acid–base titration: the reaction mixture was titrated with 0.1 N NaOH before and after 24 h of reaction using the following Equation (1):(1)$$\mathrm{Functionalization}\;\mathrm{degree}\;\left(\mathrm{mol}\%\right)=\frac{\mathrm{no}.\;\mathrm{of}\;\mathrm{mmol}\;\mathrm{reacted}\;\mathrm{COOH}}{\mathrm{no}.\;\mathrm{of}\;\mathrm{mmol}\;\mathrm{available}\;\mathrm{OH}\;\mathrm{in}\;\mathrm{PVA}}\times 100$$
where the initial PVA has 37.18 mmol/g of available OH groups [24].

### 2.3. Synthesis of 3-Mercaptopropionate Chitosan

The thiolated chitosan (CHIT-SH) was prepared through conjugation of 3-mercaptopropionic acid with chitosan. Briefly, 25 mL of 2% *w*/*v* chitosan (low Mw) in 2.5% *v*/*v* aqueous acetic acid (pH~4) was mixed with 1.2 mL of mercaptopropionic acid (13.7 mmol). Then, 2.68 g of EDC (13.7 mmol) was added to the reaction mixture, and the reaction was magnetically stirred for 6 h at room temperature. After the reaction time, the reaction mixture was added to an excess amount of ethanol (~200 mL). Then, the precipitated product was collected through centrifugation (5000 rpm × 20 min) and then was washed with ethanol several times, and it eventually dried under vacuum.

### 2.4. Preparation of PVA-CHO/CHIT-SH Hydrogels

The PVA-CHO/CHIT-SH hydrogel was prepared by simply mixing PVA-CHO solution (2, 5, and 10% *w*/*v*) in the ethanol–water system (1:1 volume ratio) with CHIT-SH solution (2, 5, and 10% *w*/*v*) in distilled water; the gelation time was based on the total concentration of polymer (2, 5, and 10% *w*/*v*) used, and the two polymers were used with the same concentration (1:1) ratios to prepare the self-assembled hydrogel at room temperature.

### 2.5. Methods of Characterization

Transmission FTIR spectroscopy was used to examine the initial polymers and modified forms. The spectra were obtained using an Avatar 330 FTIR spectrometer (Thermo Nicolet, Unicam Hungary Ltd., Budapest, Hungary) integrated with a deuterated triglycine sulfate detector and set at 400–4000 cm^−1^. The spectral resolution was adjusted to 2 cm^−1^, and 128 scans were carried out to improve the signal-to-noise ratio. The background was recorded with a pure KBr disk and was subtracted from each spectrum.

The ^1^H-NMR spectra were recorded in D_2_O and DMSO-d_6_ solutions in 5 mm tubes at room temperature (RT) using a Bruker DRX-500 spectrometer (Bruker BioSpin, Karlsruhe, Baden Württemberg, Germany) at 500 (1H) MHz with the deuterium signal of the solvent set as the lock and TMS as internal standard (1H).

Viscosity measurements were performed with a Physica MCR 301 rheometer (Anton Paar, Graz, Austria) fitted with the plate–plate type (PP20-SN13684; d = 4.5 mm), and the measurements were performed at 37 °C. Aqueous CHIT-SH and PVA-CHO solutions were prepared at different concentrations (1, 2, and 3 wt.%) and pH ranges using buffer solutions for the CHIT-SH polymer solution, and then the same volume ratios of the two solutions were mixed and left for 5 min until complete gelation after which the viscosity of the obtained hydrogel was measured directly. Additionally, the viscosities of two solutions of 2% *w*/*v* PVA-CHO and 2% *w*/*v* CHIT-SH were compared to the viscosity of 2% *w*/*v* PVA-CHO/CHIT-SH hydrogel at 37 °C to demonstrate the crosslinking of PVA-CHO with CHIT-SH through the Schiff-base mechanism which generated the hydrogel.

The solubility of chitosan and CHIT-SH was evaluated by mixing 50 mg of polymer with 10 mL of PBS solution (with different pH values: pH 6.1, pH 7.4, and pH 8.1) under strong magnetic stirring for 24 h at room temperature, and the formed suspension was then centrifuged (10,000 rpm × 30 min). The supernatant was removed, and the collected polymer was dried in the oven overnight at 50 °C and weighted to determine the soluble amount of polymer.

Ellman’s reagent was used to determine the thiol content of CHIT-SH. 4 mg of Ellman’s reagent was dissolved in 1 mL of pH = 8 phosphate buffer with 1 mM EDTA (reaction buffer) to prepare an Ellman’s reagent solution. CHIT-SH was dissolved in 0.1 N HCl solution (pH = 3.8) in the range of 0.5–1.5% *w*/*v*. Then, 2.50 mL of reaction buffer was mixed with 250 μL of CHIT-SH solution and 50 µL of Ellman’s reagent solution, and then the reaction was left for 2 h. After the reaction time, the reaction mixture was filtrated through a PVDF membrane filter (Teknokroma, Barcelona, Spain) with a pore size of 0.22 µm, and the absorbance of the filtrate was measured using a Jasco V-740 UV/Vis Spectrophotometer (ABL&E-JASCO, Vienna, Austria) at 412 nm. Calibration for thiol group content was performed using L-cysteine standards in the concentration range of 0.044–0.22 mg/mL. Each measurement was performed in triplicate; data were presented as mean ± SD.

Based on the total number of available reaction sites in the chitosan chains prior to functionalization and the results of Ellman’s method, the degree of substitution (DS%) of chitosan with 3-mercaptopropionic acid was estimated using the following (Equation (2)):(2)$$DS\%=\frac{N_{Ellman}\times 161}{W_{CHI}\times DD}\times 100$$
where N_Ellman_ is the number of thiol groups obtained by Ellman’s method (mol), W_CHI_ is the mass of chitosan (g), 161 is the average molar mass of 2-amino-2-deoxy-D-glucose units of chitosan (g·mol^−1^), DD is the degree of deacetylation of chitosan that can be calculated through ^1^H-NMR using Equation (3):(3)$$DD=100\%-\left[\frac{\left[\frac{A_{CH_{3}}}{3}\right]}{A_{H_{2}}}\times 100\%\right]$$
where *A_CH3_* is the area of methyl hydrogen of the acetyl group (acetamide) at 2.01 ppm and where *A_H2_* is the area of methyl hydrogen from the C-2 carbon of glucosamine ring at 3.1 ppm [38].

The mucoadhesive properties of the 2% *w*/*v* PVA-CHO/CHIT-SH hydrogel were investigated using a TA.XTplus Texture Analyser (Metrohm, Budapest, Hungary), and a 1 kg load cell was used. The hydrogel was fixed to the cylinder probe (with a diameter of 1 cm) using double-sided adhesive tape and was placed in contact with the pig intestinal membrane (brought from a local slaughterhouse) as in vivo mucosal surface. Five parallel measurements were conducted, and the data were presented as mean ± SD. After applying a 2500 mN preload for 3 min, the cylinder probe was pushed upwards at a predetermined speed of 2.5 mm min^−1^ to remove the disk from the actual mucosal surface. To assess the mucoadhesive behavior, the work of adhesion (mN/mm) value was estimated based on the area under the “force vs. distance” curve.

The morphology of the samples was investigated using field-emission scanning electron microscopy (SEMHitachi S-4700 microscope) with a 20 kV acceleration voltage. To validate the success of the thiolation reaction, elemental composition analysis of CHIT and CHIT-SH was done using the Röntec EDS detector at 20 keV.

Water contact angle (Θ) measurements were evaluated using drop shape analysis (EasyDrop, KRÜSS GmbH, Hamburg, Germany) device fitted with a 0.5 mm needle syringe to introduce the water drops to the surface of PVA, PVA-CHO, CHIT, CHIT-SH, and dried hydrogel (5 and 10% *w*/*v*) films at 25.0 ± 0.5 °C. Utilizing the CCD camera goniometer, the Young-Laplace equation was utilized to mathematically describe the drop contour of the captured photo using DSA100 software, and the Θ was described as the slope of the contour line at the point of contact of three phases. Thin films of samples were prepared using the solvent-casting technique, and solvent evaporation occurred in the oven at 50 °C overnight.

The swelling of lyophilized hydrogels at different concentrations was studied using gravimetric measurements. A total of 50 mg of lyophilized dry xerogel was immersed in 10 mL of PBS buffer solution at 37 °C, and the difference in the weight of swollen and nonswollen hydrogel was monitored by drying the surface of the hydrogel with a filter paper and weighing using an analytical balance at different times. The swelling ratio (g/g) was calculated using Equation (4):(4)$$\mathrm{Swelling}\;\mathrm{ratio}\;\left(g/g\right)=\frac{m_{t}-m_{o}}{m_{o}}$$
where *m_o_* and *m_t_* are the initial and swollen (at a different time) weights of the hydrogel, respectively.

The in vitro degradation of the PVA-CHO/CHIT-SH hydrogel was investigated using gravimetric measurements. Around 50 mg of lyophilized xerogel was swelled in 10 mL of PBS solution for 24 h followed by determination of the weight of the swollen hydrogel. The hydrogel was then incubated in 0.15% lysozyme (in PBS) solution at 37 °C, and the weight of the hydrogel was determined at different times using a filter paper to dry the surface water of the hydrogel before the weighting process. The degradation percentage was calculated using Equation (5):(5)$$Degredation\;\%=\frac{m_{i}-m_{t}}{m_{i}}\times 100$$
where *m_i_* and *m_t_* are the initial swollen and remaining weight of the hydrogel, respectively.

The self-healing experiment was carried out by preparing a 2% *w*/*v* PVA-CHO/CHIT-SH hydrogel in two distinct colors (blue from methylene blue dye and bright red from rhodamine b dye) and then simply combining the two hydrogels and monitoring the healing process by taking digital photos at different times (*t* = 0, 15, 60 min).

The self-assembling injectability of PVA-CHO/CHIT-SH hydrogel was investigated by preparing 7% *w*/*v* of PVA-CHO in ethanol–water (with a 1:1 volume ratio) and 7% *w*/*v* of CHIT-SH in distilled water, and then the two polymer solutions were transferred to the dual syringe. The hydrogel was prepared by depressing the syringe and immersing it in a PBS solution with pH of 7.4 at 37 °C.

Rheological characteristics of the hydrogels were analyzed using Physica MCR 302 Modular Compact Rheometer (Anton Paar, Graz, Austria). A parallel plate-type measuring device was applied (diameter of 25 mm, gap height of 0.30 mm), and measurements were carried out at 37 °C.

The healing characteristics of the gels were also examined using the continuous step strain method at 37 °C, where a low strain of 1.0% at 10 rad/s was followed by a high strain of 500% for 60 s and then reduced to 1% for recovery of hydrogel for 60 s. The step strain sweep cycle was repeated 3 times.

3D-printing application of self-assembled hydrogel was studied by mixing PVA-CHO (2% *w*/*v*, in ethanol–water, 1:1 ratio) and Chitosan-SH (2% *w*/*v*, in distilled water) with a dual syringe and writing the abbreviation of the University of Szeged (SZTE).

The in vitro experiments for the tilorone dihydrochloride release were performed using a cellulose membrane (Sigma–Aldrich, avg. flat width of 25 mm (1.0 in.)). The hydrogel was prepared by mixing 1 mL of PVA-CHO at the desired concentration (2, 5, and 10% *w*/*v*) in an ethanol–water (1:1 ratio) system and 1 mL of CHIT-SH at the desired concentration (2, 5, and 10% *w*/*v*) in distilled water with an appropriate tilorone drug amount (2.3 mg) and leaving it for 5 min to complete the gelation process. The amount of tilorone released from the pure drug form and hydrogel forms at different concentrations (2, 5, and 10% *w*/*v*) were spectrophotometrically quantified by measuring the difference in intensity of the characteristic tilorone absorbance at λ = 270 nm. According to the calibration curve that was previously estimated to be within the range of 0–0.024 mg/mL of tilorone solution, tilorone concentration was directly related to the maximum absorption values observed at this wavelength (c_Tilorone_ (mg/mL) = (A_270 nm_)/150; R^2^ = 0.9997). Solid powder of pure tilorone drug and hydrogel forms (equal to 0.023 mg/mL of tilorone; the same drug amount in all the experiments) were placed in a cellulose membrane that was closed and immersed in 100 mL of a PBS buffer solution at 37 °C (pH = 7.4; 0.9 wt.% NaCl content). During the release experiment, 3 mL of the aqueous buffer solution was drawn from the dissolution medium at predefined time intervals, and the average tilorone concentration and standard deviations were calculated (n = 3). The in vitro release of tilorone dihydrochloride was also studied at 37 °C in the presence of 0.15% lysozyme (in PBS) solution. The same drug content was used in the PBS solution only, but 1 mL of aqueous buffer solution was taken during the release experiment and diluted with 2 mL of PBS solution to be able to measure the concentration of tilorone because the presence of lysozymes disturbs the baseline during the measurement of absorbance. The concentration of released drug was calculated based on the absorbance multiplied by the dilution factor, and the measurements were repeated three times with calculations of the standard deviation.

The cytotoxicity test was performed in 96-well cell culture microplates with 3-(4,5-dimethylthiazol-2-yl)-2,5-diphenyltetrazolium bromide (MTT) assay. MRC-5 (human embryonic lung fibroblasts) cells were seeded at a density of 1 × 10^4^ cells/well and incubated for 24 h at 37 °C. Next, the disc-shaped hydrogel samples (5 and 10%) were cut to the appropriate size (cut to size to fit the wells) and were immersed in the cell suspension except for the medium control wells (MC). After a 24 h incubation period at 37 °C, each well was injected with 20 μL of (5 mg/mL) MTT solution (Sigma–Aldrich, Germany). Following a 4 h incubation at 37 °C, 100 μL of 10% sodium dodecyl sulfate (SDS, Sigma–Aldrich) was poured into each well for dissolution of the formed formazan, and then the optical density (OD) was recorded after 24 h. In the experiment, the solvent had no influence on cell growth at the doses utilized in cytotoxicity test compared to cell control (CC).

### 2.6. Statistical Analysis of the Results

The significance of the results was determined with a two-sample t-test. A phenomenon was considered significant if *p* < 0.05, which was denoted with * asterisk. In the case of *p* < 0.01, it was denoted with ** asterisks, and, if *p* < 0.001, it was marked with *** asterisks.

## 3. Results and Discussion

### 3.1. Structural Characterization of the Modified Polymers and Hydrogel

Although CHIT has a high biocompatibility and biodegradability [10,11], it is not suitable for use in the preparation of injectable hydrogels due to limitations such as poor solubility at a physiological pH and low mucoadhesive properties. Thus, CHIT was modified to improve its water solubility and mucoadhesive characteristics. PVA with pendant OH has high mechanical, biocompatibility, and biodegradability properties, and its OH groups can be easily converted to benzaldehyde derivatives, which can then be employed as crosslinks for the amino groups of modified CHIT to generate the dynamic hydrogel as shown in Figure 1. Additionally, the measured (by titration) functionalization degree was 5.9 mol% (out of 10.7% of the nominal functionalization degree). The existence of the available CHO (5.9 mol%) pendant groups in the PVA backbone offers an adequate crosslinking agent for the NH_2_ group of CHIT through the formation of a pH-dependent Schiff base (imine bond), with a higher pH (≥5; OH^–^) promoting the bond formation and a lower pH (≤4; H^+^) inducing bond cleavage [6]. To increase the mucoadhesive properties of the formed hydrogel, CHIT was reacted with 3-mercaptopropionic acid using EDC as a coupling agent (see Figure 1), forming CHIT with a free thiol terminal (thiol content = 201.85 ± 12 µmol/g) capable of forming a disulfide bond with the cysteine-rich subdomains of the mucus layer and resulting in an increased hydrogel residence time and improved bioavailability [39].

The successful modification of the initial polymers was confirmed by the FTIR measurements as shown in Figure 2. The main characteristic peaks of PVA were 3280 cm^−1^ (O–H stretching vibration), 2960 cm^−1^ (CH_2_ asymmetric stretching vibration), 2925 cm^−1^ (CH_2_ symmetric stretching vibration), 1735 cm^−1^ (C=O carbonyl stretching vibration), 1425 cm^−1^ (C–H bending vibration of CH_2_), 1380 cm^−1^ (C–H deformation vibration), 1325 cm^−1^ (CH_2_ wagging vibration), 1245 cm^−1^ (C–O–C stretching vibration), 1100 cm^−1^ (C–O stretching of acetyl groups), and 840 cm^−1^ (C–C stretching vibration) [24,40,41]. The modification reaction was confirmed by the appearance of the C=O stretching vibration of the aldehyde at 1705 cm^–1^ which overlapped with the ester (C=O) carbonyl at 1730 cm^–1^. The formation of an intramolecular hydrogen bond between the CHO and OH groups of PVA led to an increase in the intensity and the wavenumber shifting of the unmodified OH of the PVA backbone [24,42]. The peak at 2860 cm^–1^ was related to the C–H stretching vibration of the aldehyde group [24]. The peaks between 1650 and 1505 cm^−1^ were due to the C=C stretching of the aromatic ring, while the peak at 1210 cm^−1^ was due to the C–O ester asymmetric stretching vibration band. Both of the peaks at 820 cm^−1^ and 760 cm^−1^ were due to the C–O–C deformation vibration and the out-of-plane =C–H bending vibration on the aromatic ring, respectively [43].

Furthermore, the successful substitution reaction was also confirmed with UV- absorption spectroscopy by the appearance of the absorption band of a benzaldehyde moiety at a maximum wavelength λ_max_ = 255 nm [44] (see Appendix A).

The main characteristic peaks of the initial chitosan (CHIT) were detected at 3440, 2890, 1670, 1565, 1430, 1380, 1160, 1080, and 1035 cm^−1^. These peaks were attributed to the stretching vibrations of the –OH and –NH_2_ groups, C–H stretching vibration, C=O stretching of the amide I band, bending vibrations of the N–H (N-acetylated residues, amide II band), C–H bending, O–H bending, antisymmetric stretching of the (C–O–C) bridge, and skeletal vibration involving C–O stretching [45,46,47]. The successful substitution reaction was confirmed by the dimension of the peak at 3440 cm^−1^ that was assigned to the –OH and –NH_2_ groups and by the appearance of a new peak due to the stretching vibration of the C–H of the mercaptopropionate moiety at 2975 cm^−1^, and the broad peak at 2570 cm^−1^ was related to the S–H group, which confirmed the grafting of 3-mercaptopropionic acid to chitosan. The new peak at 1240 cm^–1^ was due to C–SH stretching [48,49]. There was a shifting of the peak of the amide I band with a decrease of around 30 cm^−1^ in addition to the appearance of a new peak near the amide II band at 1545 cm^−1^ that could be attributed to the conjugation of mercaptopropionic acid to the NH_2_ of chitosan [48,49]. A new peak appeared in the FTIR spectrum of the 10% *w*/*v* PVA-CHO/CHIT-SH hydrogel at 1647 cm^−1^, indicating the formation of a Schiff base (–CH=N–) [50] and confirming the interaction between the NH_2_ groups of CHIT–SH and the CHO groups of PVA-CHO for the formation of the dynamic hydrogel.

After the chemical characterization of the polymers with FTIR, the ^1^H NMR spectra of the initial polymers and modified forms were also recorded (see Appendix A). The PVA-CHO spectrum showed characteristic PVA peaks with diminished hydroxyl peaks, 2-2 aromatic protons, and a characteristic CHO signal at a higher ppm (10.11 ppm; Appendix A). The CHIT-SH spectrum also showed the characteristic peaks of chitosan, while the peaks of two CH_2_ groups also appeared that we hypothesized belonged to the 3-mercaptopropionate moiety (Appendix A). We hypothesized that these new peaks and the results of the further analysis showed functionalization in both polymers. According to the ^1^H NMR measurements, the DD of the initial chitosan was equal to 75.02%, and the DS% of chitosan with 3-mercaptopropionic acid was 4.43%. The typical yields based on the collected polymers and the degree of modification (or substitution) of the PVA-CHO and CHIT-SH products were ~91% and ~84%, respectively.

At a physiological pH, one molecule of an aldehyde and two molecules of an alcohol could not form an acetal bond; hence, the reaction needed anhydrous acid catalysis for completion since a late stage in the mechanism entails the removal of a hydroxide (which, under acidic conditions, manifests as water) from a tetrahedral intermediate [51]. However, the hydroxyl and aldehyde moieties of PVA-CHO could be masked as hemiacetal at a physiological pH, which resulted in the biocompatibility of PVA-CHO with a variety of cellular strains since the reactive aldehyde moiety is often cytotoxic [23,52]. Because the hemiacetal was unstable and reverted to the original aldehyde and alcohol, the reaction of hemiacetal formation was presented as an equilibrium. A thioacetal was another suggested bond formation between the SH groups and CHO groups, and the mechanism of the reaction was similar to acetal formation and required the presence of acid catalysis, such as that of Lewis acids (BF_3_ or ZnCl_2_) in an ether solvent; thus, the formation of an acetal or thioacetal could not form under a physiological pH in the absence of an acid catalyst [51].

The formation of a disulfide bond is fundamentally based on the local environment and involves a reaction between the two sulfhydryl (SH) groups, where the attack of the nucleophile (S^−^) of one sulfhydryl group on another SH group releases an electron to create a disulfide bond. The formation of a disulfide bond is directly affected by two major factors: (1) the difference between the pKa of the implicated thiol groups and the pH of the local environment (a lower pH limiting reactivity and a higher pH encouraging enhanced reactivity) and (2) the redox environment (with lower reactivity under more reducing conditions and greater reactivity under more oxidizing conditions) [53]. The disulfide bond was a favor to form under physiological conditions (pH 7.4) and was one possible bond formation in our hydrogel system; however, it was not the main bond because our hydrogel required less than 1 min for its formation compared to the low rate of the disulfide bond formation [54]. Furthermore, the mucoadhesive results demonstrated the existence of free thiol groups, which were responsible for the mucoadhesive characteristic of our hydrogel as we explain later. Appendix A shows some of the possible bond formations in the hydrogel system based on the acetal and disulfide bonds; however, none of these showed gel formation, confirming our interpretation of hydrogel formation being based on an imine bond.

### 3.2. Physicochemical Characterization and pH-Dependent Sol–Gel Transition of PVA-CHO/CHIT-SH Hydrogel

According to our hypothesis, mixing the PVA-CHO solution with the CHIT-SH polymer resulted in the formation of the hydrogel at a neutral pH due to the formation of a Schiff base (imine bond), so the viscosity of the 2 and 3% *w*/*v* PVA-CHO/CHIT-SH hydrogels at different pH values was investigated using a different buffer solution as the solvation medium. Figure 3A shows the comparison of the viscosity of the 2% *w*/*v* polymer solutions (PVA-CHO and CHIT-SH) and the mixtures at the same concentration (2% *w*/*v*). As can be seen, the formation of the hydrogel due to the Schiff-base reaction between the NH_2_ groups of CHIT-SH and the CHO groups of PVA-CHO resulted in a significant increase in the viscosity that was more than one thousand times greater than that of the used polymer solutions, which had viscosities of 4.6 and 5.8 mPa·s, respectively. Figure 3B shows the viscosity of the 2 and 3% *w*/*v* PVA-CHO/CHIT-SH hydrogels as a function of the change in pH. The viscosity values of the two prepared hydrogels were greater at pHs of 5.3 and 7.2, reaching around 10,000 mPa·s for 2% *w*/*v* PVA-CHO and more than 20,000 mPa·s for 3% *w*/*v* PVA-CHO, respectively; however, at an acidic pH, the viscosity was low due to the breakdown or disintegration of the imine bond as shown in Figure 3B. The inserted photos of the hydrogel (gel) at a neutral pH and the low viscosity solution (sol) at an acidic pH are shown in Figure 3B as well as the mechanism of hydrogel formation. The disulfide bond formation depended on the environmental pH, with a lower pH limiting reactivity and leading to the presence of more thiol and a higher pH favoring increased reactivity in order to form a disulfide bond [53,54]. Appendix A shows the pH-dependent sol–gel transition of the PVA-CHO/CHIT-SH hydrogel in the presence of the tilorone drug (0.023 mg/mL). It was seen that the sol–gel transition also took place in the presence of the drug. In other words, PVA-CHO/CHIT-SH was in the “sol” form when in an acidic medium and in the “gel” form when in a neutral one. Moreover, this increase in viscosity could aid the use of the PVA-CHO/CHIT-SH hydrogel, with its biocompatible character, in biomedical applications at a physiological pH, such as tissue engineering and drug delivery as seen later.

Chitosan had poor water solubility at a pH > 6, where it lost its positive-charge density and induced the formation of aggregates and precipitates [55]. Appendix A compares the solubility of the initial chitosan and CHIT-SH in different pH buffer solutions. After being modified with 3-mercaptoproionic acid (SD = 4.43%), chitosan became more soluble in basic and neutral media. CHIT-SH exhibited high solubility values of around 4.97 mg/mL (99% solubility; absolute value = 5 mg/mL) and 3.89 mg/mL (77% solubility) at a pH of 7.4 and a pH of 8.1, respectively, in contrast to the initial chitosan that showed a very low solubility at the same pH values. At a pH of 6.1, both chitosan and CHIT-SH had a low solubility; however, CHIT-SH had a higher solubility than chitosan. The increased solubility in the case of CHIT-SH was due to the presence of the hydrophilic functional groups of SH that interrupted the intermolecular H-bond and crystalline structures of chitosan, resulting in a significant enhancement in solubility.

Next, the hydrophilicity and hydrophobicity of the dried polymer films prepared from the hydrogels at different (5 and 10% *w*/*v*) concentrations were investigated by measuring their water contact angles and comparing them to the PVA-CHO and CHIT-SH films, which were all prepared using the solvent-casting technique. The modification of the initial hydrophilic PVA (55 ± 1°) with 4-formylbenzoic acid led to an increase in the hydrophobicity of PVA-CHO (91 ± 3°), while the modification of the initial CHIT (73 ± 1°) with 3-mercaptopropionic acid led to an increase in the water solubility and hydrophilicity of CHIT-SH (45 ± 2°) as shown in Figure 4. Thus, the mixing of the hydrophilic (45 ± 2°) CHIT-SH with the hydrophobic (91 ± 3°) PVA-CHO led to the formation of a crosslinking film with moderate hydrophilic/hydrophobic characteristics (75 ± 2° for 10% *w*/*v* PVA-CHO/CHIT-SH and 65 ± 3° for 5% *w*/*v* PVA-CHO/CHIT-SH) as shown in Figure 4. Increasing the polymer concentration resulted in an increase in the hydrophobicity of the formed hydrogel due to an increase in the crosslinking (imine bond) network structure, which could be a suitable strategy for controlling the drug release from the hydrogel matrix as seen in the in vitro drug-releasing section.

After studying the wetting properties of the dry polymers, the swelling of the lyophilized hydrogels at different (2, 5, and 10% *w*/*v*) concentrations was also studied under physiological conditions (PBS, pH 7.4, 37 °C) as shown in Figure 5A. Increasing the crosslink density through the formation of an imine bond between the PVA-CHO and CHIT-SH polymers led to the restriction of the swelling of the hydrogel in the PBS buffer solution as can be seen in Figure 5A. The 10% *w*/*v* PVA-CHO/CHIT-SH had the lowest equilibrium swelling (1.7 g/g) ratio, while the 2% *w*/*v* PVA-CHO/CHIT-SH had the highest swelling (2.9 g/g) ratio. Due to its moderate hydrophilic/hydrophobic characteristic (see Figure 4), the water uptake of the dry xerogel increased in a short time and then remained constant. The degradation of the PVA-CHO/CHIT-SH hydrogels at different concentrations was investigated using 0.15% lysozyme/PBS buffer solution at 37 °C as shown in Figure 5B. The increase in the crosslinking structure of the hydrogel matrix was accompanied by an increase in the hydrolysis of the imine bond as well as the enzymatic hydrolysis of the glycosidic bonds of the acetyl residues in chitosan by lysozymes, which was why the rate of hydrolysis and degradation was higher in the case of 10% *w*/*v* PVA-CHO/CHIT-SH compared to 2% *w*/*v* PVA-CHO/CHIT-SH, with a lower degradation rate during the first 15 days of the experiment. However, the last 5 days of the experiment showed a steady rise in the rate of degradation with a very small difference in the final degradation (~52−55%) percentage. The hydrogel was a mixture of chitosan (that could easily be degraded with lysozymes) and PVA with a 1:1 mass ratio, and, after 20 days, the maximum degradation was about 55%, which was still acceptable if we assumed that only chitosan hydrolysis occurred. The prepared PVA-CHO/CHIT-SH hydrogels at different concentrations had the appropriate physicochemical properties that recommended its use for local drug delivery and tissue engineering applications as seen later.

### 3.3. Mucoadhesive Properties of PVA-CHO/CHIT-SH Hydrogel

Thiolated macromolecules were one of the most efficient mucoadhesive polymers due to their ease in forming disulfide bonds (covalent bonds) with the cysteine-rich subdomains of the mucus gel layer through the thiol–disulfide exchange mechanism in addition to the van der Waals’ forces and hydrogen bonds that led to an increase in the residence time of the polymers inside the body and that improved bioavailability [39,40]. In order to provide free thiol groups for the generated hydrogel, the 3-mercaptopropionate moiety was used to modify the NH_2_ groups of CHIT via amide bond formation. These free terminal thiol groups enabled the hydrogel to be able to form covalent S-S bonds with the SH groups of the mucous membrane [39,40]. The thiol content of CHIT-SH was evaluated using Ellman’s reagent method, with a result of 201.85 ± 12 µmol/g, as well as the EDX measurement, showing the presence of a high sulfur content due to the thiol group of the 3-mercaptopropionate moiety of CHIT-SH as shown in Figure 6 compared to pure CHIT, which exhibited no sign of the existence of the sulfur element.

The two significant limitations of using pure CHIT as a mucoadhesive polymer were that it had a low mucoadhesive strength and a low water solubility at neutral and basic pHs [56]. As a result, the work of adhesion of the PVA-CHO/CHIT-SH hydrogel (CHIT-SH with a thiol group) was compared with the reference R- PVA-CHO/CHIT hydrogel (pure CHIT without a thiol group) by using a pig intestinal membrane as a mucous layer. As the work of adhesion of the thiol-containing polymer that was in contact with the pig intestinal tissue was measured to be significantly higher (*p* < 0.05) than that of the reference, it could be considered more mucoadhesive (Figure 7). This phenomenon could be attributed to the formation of disulfide bonds between the mucus and the polymer via the thiol–disulfide exchange mechanism. The results indicated a prolonged residence time on the mucosa compared to the PVA-CHO/CHIT mixture.

Several studies have reported using benzaldehyde derivatives in the production of dynamic hydrogels based on chitosan [6,9,57,58,59], but in our report, we introduced a new concept for the injectable hydrogel by increasing its mucoadhesive properties as shown in Figure 7 as well as by increasing the water solubility of chitosan as shown in Appendix A, which improved the shortcomings of the chitosan and raised the interest in using it in neutral media and physiological conditions without the need for an acid.

### 3.4. Self-Assembled and Self-Healing Hydrogel Formation

For biomedical applications, in situ forming hydrogels are preferable over prefabricated hydrogels as there is no need for surgical operations because their gelation can happen under physiological circumstances upon injection. Additionally, the hydrogel can include biological components and drug particles by simply mixing them with the precursor polymer solution due to the precursor solution’s original fluid character, which guarantees correct shape adaption [60]. The injectable hydrogel was crosslinked via a Schiff-base reaction between the amino (NH_2_) groups on CHIT-SH and the aldehyde (CHO) groups on PVA-CHO. Figure 8A shows the applicability of the injection of the 7% *w*/*v* hydrogel into the PBS solution at 37 °C. Both polymers were pushed at the same speed ratio, and a mixing procedure took place in the vein that collected the exits of both syringes in order to form the PVA-CHO/CHIT-SH hydrogel with PBS buffer stability. The original color of the PVA-CHO/CHIT-SH hydrogel was pale white as shown in Figure 3, and methylene blue dye was used to exhibit the formation of the hydrogel in the PBS solution and to confirm the encapsulating feature of the formed gel structure as shown in Figure 8A. A vial inversion test was performed to measure the gelation times (see Figure 8B), which ranged from 45 to 7 s based on the concentration of the polymers used (2, 5, and 10%, respectively).

In addition to in situ gel formation, the PVA-CHO/CHIT-SH hydrogel demonstrated self-healing characteristics, which were attributed to the dynamic of Schiff-base production under certain circumstances. Figure 8C shows the self-healing of two 2% *w*/*v* PVA-CHO/CHIT-SH hydrogels with two different colors due to the use of methylene blue dye and rhodamine B dyes; two half hydrogels were attached and left to heal for 1 h. After the healing time, the two half hydrogels were completely attached as one whole, and the formed hydrogel was dried in the oven at 60 °C overnight. The dried layer of the PVA-CHO/CHIT-SH hydrogel showed durability against the detachment force as presented in Figure 8C.

Rheological recovery experiments were carried out using the continuous step strain method to assess the elastic response and self-healing ability of the PVA-CHO/CHIT-SH hydrogel by applying a strong deforming shear and then allowing it to recover. The shear used for the 2, 5, and 10% *w*/*v* PVA-CHO/CHIT-SH hydrogels was 500% at a fixed angular frequency of 10 rad/s at 37 °C and indicated the gel–sol transition strain for our gel system, which was determined through oscillatory rheology measurements as shown in Appendix A. Figure 9 shows the changes in the strain-dependent modulus between the three hydrogels as well as the changes in the sol–gel transition (damage healing) of the hydrogels as a result of applying a large oscillatory amplitude (strain = 500%) strain and having a recovery at a low strain (strain = 1%). When the strain was increased from 1 to 500%, the storage modulus (G′) value of the 10% PVA-CHO/CHIT-SH hydrogel suddenly decreased from ∼2090 Pa to ∼30 Pa and then recovered to ∼1700 Pa by returning the strain to 1% (Figure 9A). Additionally, the 5% PVA-CHO/CHIT-SH hydrogel showed a self-healing characteristic similar to that of the 10% PVA-CHO/CHIT-SH hydrogel, where the G′ value decreased from ∼450 to ∼20 Pa due to increasing the stain to 500% and recovered to ∼440 Pa as shown in Figure 9B, and the 2% PVA-CHO/CHIT-SH hydrogel had a self-healing property similar to those of the above-mentioned hydrogels as shown in Figure 9C. Due to an increase in the crosslinking density of the hydrogel structure as a result of increasing the polymers’ concentration, the G′ showed a direct relationship with the increasing hydrogel concentration. The hydrogel’s breaking and healing characteristics could be alternately repeated several times, and, also, the inner hydrogel network recovered quickly, demonstrating the rapid and highly efficient self-healing ability of our dynamic (Schiff-base) hydrogels.

The injectable hydrogel was also intended to be used in tissue engineering [61,62]. Unfortunately, in comparison to solid-state polymeric scaffolds, such as poly(lactic acid) (PLA) and polycaprolactone (PCL), hydrogels seem to have an “irreparable” disadvantage in mechanical characteristics. Nonetheless, the benefits of hydrogels, particularly for in situ gel formation, are irreplaceable, implying that injectable hydrogels can operate not only as scaffolds for supporting reasons but also as functional implants for the control of tissue generation [61,62]. Based on calcium-crosslinked alginate, Poldervaart et al. produced a hydrogel scaffold for bone tissue engineering [63]. With the same approach, PVA-CHO/CHIT-SH can be used for to induce muscle hypertrophy by incorporating the BMP inducer tilorone in order to increase the muscle mass of the atrophied vocal fold. Tilorone induced BMP signaling and increased BMP-7 expression [32]. Appendix A shows the applicability of the PVA-CHO/CHIT-SH hydrogel as bioink by writing the abbreviation for the University of Szeged.

Some materials have the intrinsic ability to repair themselves after suffering damage, and this property is known as self-healing. Reversible interactions, which can be broken and then independently re-established, are what give live tissues this ability [64]. Due to the dynamic equilibrium between the Schiff-base linkage and the aldehyde and amine reactants, our PVA-CHO/CHIT-SH hydrogel had self-healing properties as shown in Figure 9. Due to injection through thin needles and retention at target sites, hydrogels with self-healing characteristics are excellent candidates for use in drug/cell delivery applications and 3D printing in order to protect the drug/cell and provide a prolonged releasing process.

### 3.5. In Vitro Drug-Release Measurements

The topical delivery of drugs using injectable hydrogels is preferable to systemic administration due to goals of prolonged release and biodistribution control [65,66]. Hydrogels can have a high loading of water-soluble drugs due to their in situ generated 3D network and water dispersion phase [2]. Furthermore, they have the ability to encapsulate drugs that are insoluble in water [67]. Additionally, the term “drugs” includes macromolecular therapeutics and bioactive compounds such as proteins, peptides, genes, and even cell organs and living cells [2,68,69,70]. Drug release from a hydrogel matrix, on the other hand, might be influenced by a variety of mechanisms [70,71]. The release rate of injectable hydrogel formulations may be altered significantly by external stimuli, whereas crosslinking reactions or mechanisms are reversible; moreover, hydrogels may incorporate environmentally sensitive moieties in their building ingredients.

Tilorone as an antimuscle atrophy drug was encapsulated in PVA-CHO/CHIT-SH hydrogels at different (2, 5, and 10% *w*/*v*) concentrations, and the release of tilorone from different compounds is shown in Figure 10; its release from the pure form of tilorone was too quick (56.8 h^−1^ based on the initial slope of the curve at 1.5 h), with a maximum releasing amount of 98% (0.023 mg/mL), while tilorone’s release from the hydrogel networks was prolonged (31.6 h^−1^, 17.9 h^−1^, and 12.7 h^−1^ for 2, 5, and 10% *w*/*v* PVA-CHO/CHIT-SH, respectively, calculated based on the initial slope of the curve at 1.5 h) and crosslinking was dependent. After 7 h of the releasing process under physiological conditions (37 °C, pH 7.4), the percent of drug release from the 2% *w*/*v* PVA-CHO/CHIT-SH hydrogel was about 81% (0.0187 mg/mL), whereas the percent of drug release from the 10% *w*/*v* PVA-CHO/CHIT-SH hydrogel was 50% (0.0114 mg/mL). Increased polymer and hydrogel concentrations resulted in an increased crosslinking density, which can prolong the diffusion of the water-soluble drug from the formed hydrogel matrix as shown in Figure 10B, which displays the decrease in the maximum concentration of the tilorone obtained from the releasing experiment as a function of the increase of the hydrogel concentration.

The release of tilorone in the presence of lysozymes in a PBS solution (0.15% lysozyme/PBS buffer solution) at 37 °C was also investigated as shown in Figure 10C; the release of the drug from the hydrogel structure increased by 1–6%, based on the crosslinking density, compared to its release in a pure PBS buffer solution, and the reason for this was the enzymes’ degradation of the hydrogel structure, which helped to increase the drug’s release.

The kinetics of drug release in the case of the absence of lysozymes was interpreted using the different kinetic models shown in Table 1. The release of pure tilorone was fitted with the first-order model, which explained that the release kinetics were dependent on the concentration of the drug in the dosage form [72], while the release from the hydrogel form at different concentrations was fitted with the Higuchi release model, which suggested that the drug was released through diffusion from the hydrogel network (matrix system) [72]. Additionally, it can be seen that the KH (Higuchi release constant) decreased with an increase in the concentration of the hydrogel due to an increase in the crosslinking density, which impeded the diffusion of the drug from the hydrogel network.

Furthermore, the kinetics of the drug release in the presence of lysozymes was investigated using the same kinetic models in Table 2. The release of pure tilorone was fitted with the first-order model, and, also, the release from the hydrogel forms was fitted with the Higuchi release model, with a slight increase in the release rate compared to the in vitro release in the absence of the enzymes; this phenomenon was due to the degradation of the hydrogel (~5% during the release experiments) in the presence of the enzymes as shown in Figure 5. Although the release rate of the drug from the hydrogel forms increased, the hydrogel could prolong the release as the crosslinking/concentration of the hydrogel rose.

The cytotoxicity of the hydrogel was evaluated using MRC-5 (human embryonic lung fibroblasts) cells. The results clearly demonstrated that the hydrogels were nontoxic (Appendix A). It was also obvious that the amount of cells (cell proliferation) increased more in the case of the hydrogel samples (OD = 0.289 ± 0.036 for PVA-CHIT 5%, and OD = 0.273 ± 0.031 for PVA-CHIT 10%) compared to the control (OD = 0.14 ± 0.028) due to the enhanced growth of the cells on the solid surface of the hydrogel (Appendix A). A similar effect was already reported in the case of other biocompatible hydrogels [73,74].

## 4. Conclusions

A novel class of CHIT/PVA-based in situ gel-forming hydrogels was produced, which were prepared by mixing PVA-CHO and CHIT-SH aqueous solutions to form dynamic Schiff-base bonds. The chemical structures of the modified polymers and the formed hydrogels were confirmed through FTIR spectroscopy measurements. Rheological studies and visual experiments both demonstrated the injectability and self-healing characteristics of this class of hydrogels. The mucoadhesive properties of the produced hydrogels were evaluated by measuring the work of adhesion to the pig intestinal membrane, which was shown to be substantially greater (*p* < 0.05) than that of the reference which was without thiol groups. The release of tilorone from the hydrogel was based on the crosslinking density/concentration of the hydrogel; the 10% *w*/*v* PVA-CHO/CHIT-SH hydrogel had the slowest releasing (21.7 h^−1/2^) rate and the maximum releasing concentration (0.0114 mg/mL) compared to the 2% *w*/*v* PVA-CHO/CHIT-SH hydrogel which had a fast releasing rate (34.6 h^−1/2^) and the maximum releasing concentration (0.0187 mg/mL). Given the features of this type of hydrogel, applications such as injectable and in situ forming drug delivery systems, wound dressings for skin injuries, and tissue regeneration scaffolds are being considered. Hopefully, the produced hydrogel will be effective in treating local fibrosis or muscle atrophy.

## Figures and Tables

**Figure 1 pharmaceutics-14-02723-f001:**
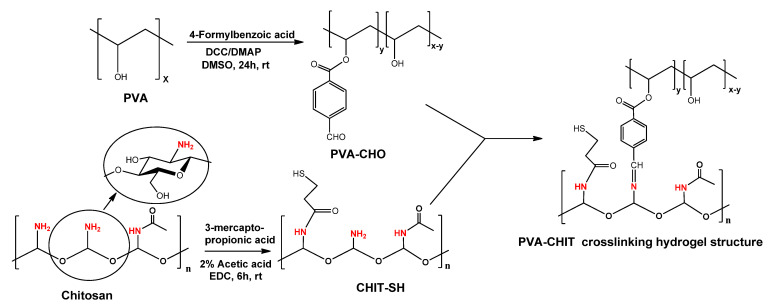
Scheme of synthesis of PVA-CHO and CHIT-SH as well as the crosslinking structure formation.

**Figure 2 pharmaceutics-14-02723-f002:**
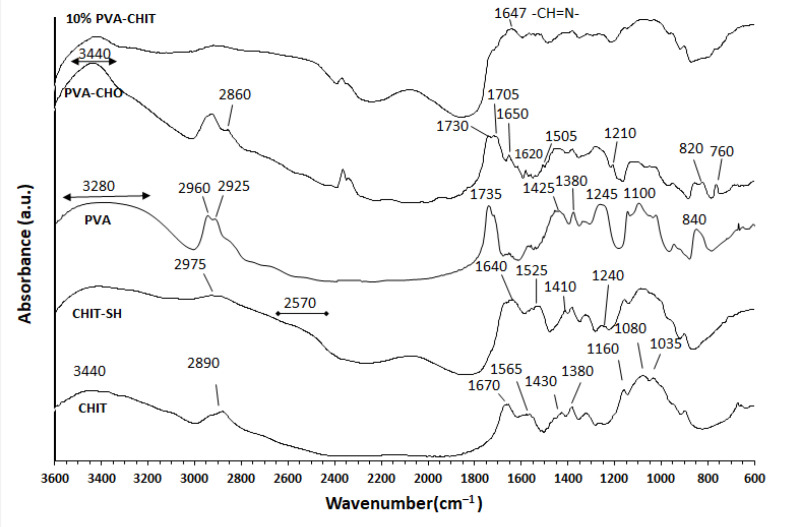
FTIR spectra of initial PVA, 4-formyl benzoate-PVA (PVA-CHO), initial CHIT, 3-mercaptopropionate CHIT (CHIT-SH), and PVA-CHO/CHIT-SH hydrogel.

**Figure 3 pharmaceutics-14-02723-f003:**
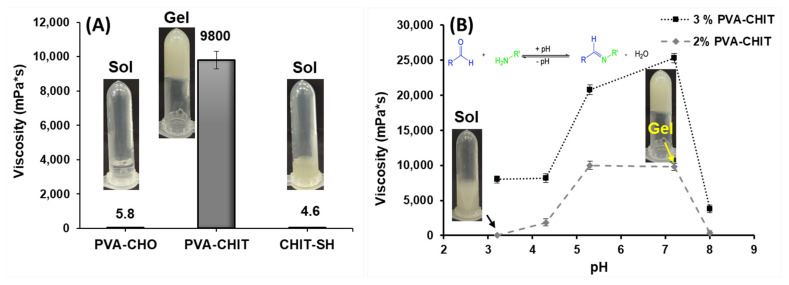
Viscosity measurements of initial polymers (2% *w*/*v*) and prepared hydrogel (2% *w*/*v*) at 37 °C (**A**), and the viscosity measurements of 2 and 3% *w*/*v* PVA-CHO/CHIT-SH hydrogels at 37 °C as a function of pH (**B**).

**Figure 4 pharmaceutics-14-02723-f004:**
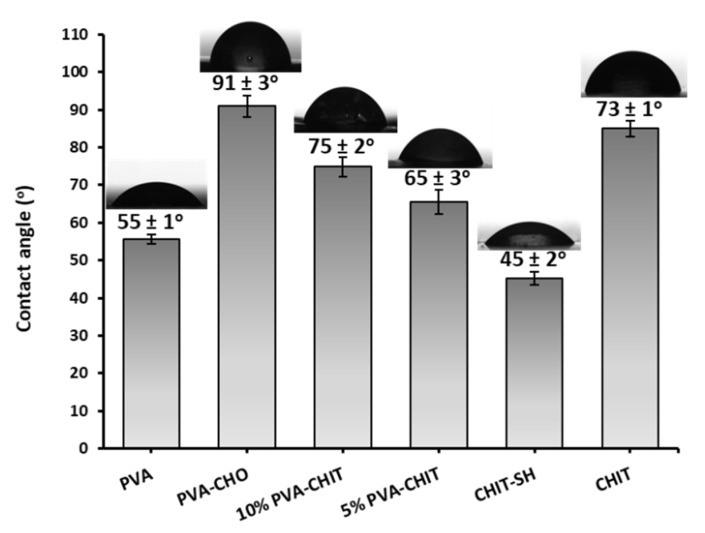
The contact angle values for modified polymers and crosslinking PVA-CHO/CHIT-SH.

**Figure 5 pharmaceutics-14-02723-f005:**
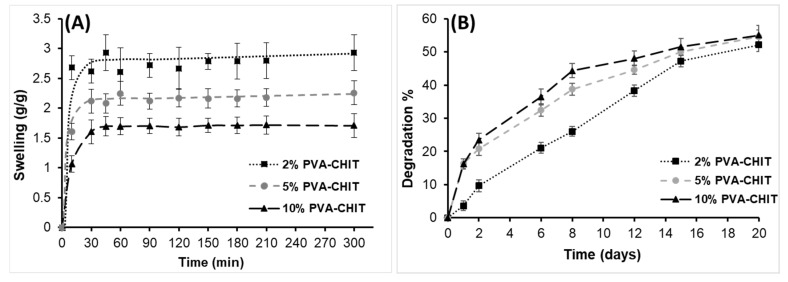
The swelling (g/g) ratio of the PVA-CHO/CHIT-SH hydrogels at different (2, 5, and 10 *w*/*v*%) concentrations in PBS buffer solution at 37 °C (**A**). Degradation of hydrogels at different (2, 5, and 10 *w*/*v*%) concentrations in 0.15% lysozyme/PBS buffer solution at 37 °C (**B**).

**Figure 6 pharmaceutics-14-02723-f006:**
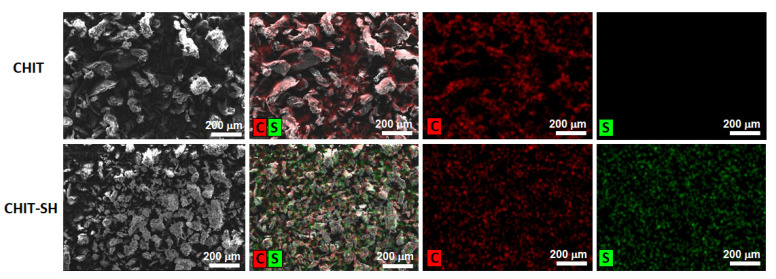
EDX measurements of CHIT and CHIT-SH, C and S are the carbon and sulfur elements, respectively.

**Figure 7 pharmaceutics-14-02723-f007:**
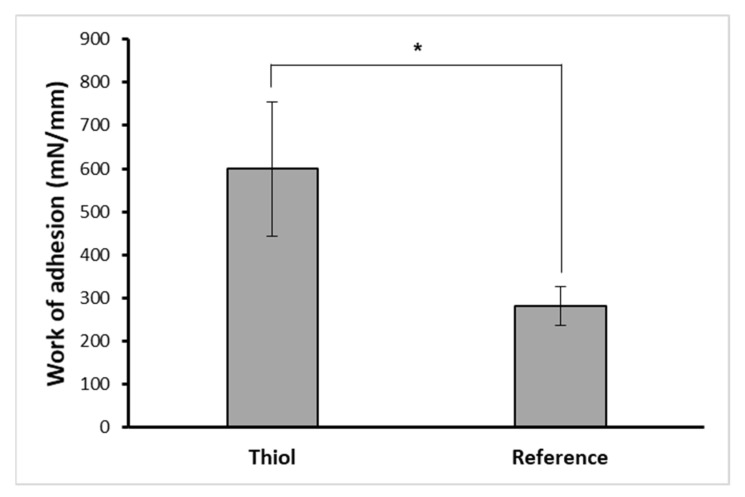
The work of adhesion of mucoadhesive thiolated hydrogel (2% *w*/*v*) and reference hydrogel (2% *w*/*v*) using pig intestinal membrane, * asterisk means significant difference (*p* < 0.05).

**Figure 8 pharmaceutics-14-02723-f008:**
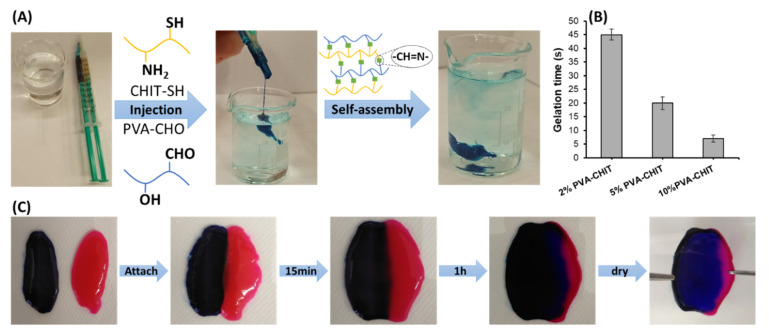
(**A**)The self-assembly of injectable hydrogel (7% *w*/*v*) with methylene blue dye in PBS buffer solution (pH 7.4), (**B**) the gelation time of the PVA-CHO/CHIT-SH hydrogel as a function of hydrogel concentration, and (**C**) the self-healing properties of the prepared hydrogel (2% *w*/*v*).

**Figure 9 pharmaceutics-14-02723-f009:**
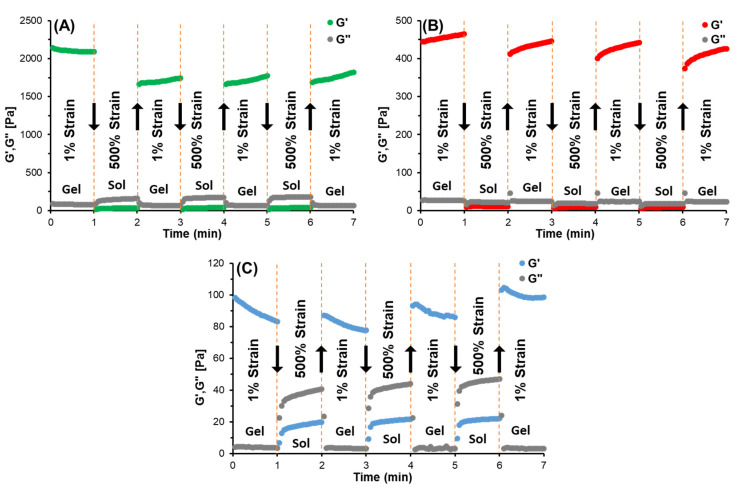
The self-healing and recovery of a PVA-CHO/CHIT-SH hydrogel after a high shear load through a continuous (1% strain → 500% strain → 1% strain) step strain method (**A**) 10% *w*/*v* PVA-CHO/CHIT-SH hydrogel, (**B**) 5% *w*/*v* PVA-CHO/CHIT-SH hydrogel, and (**C**) 2% *w*/*v* PVA-CHO/CHIT-SH hydrogel. The arrow indicates the change in the storage modulus (G′).

**Figure 10 pharmaceutics-14-02723-f010:**
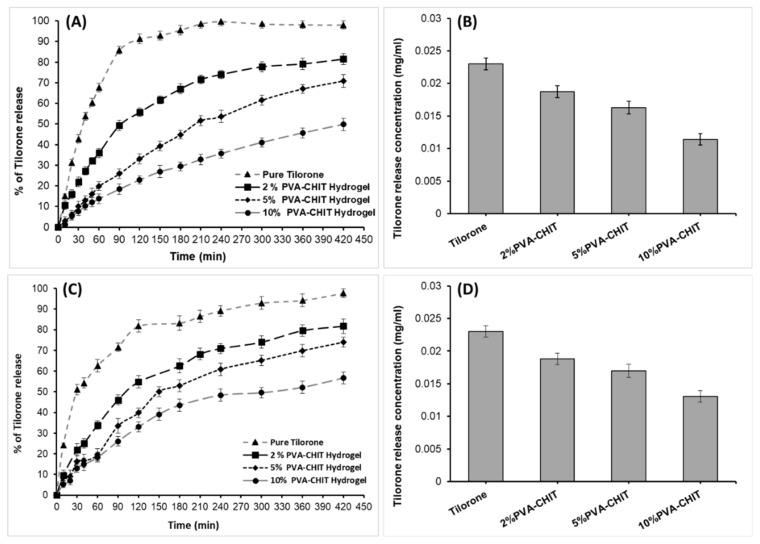
(**A**) In vitro release of tilorone dihydrochloride from its pure form and from different hydrogel (2, 5, and 10% *w*/*v*) concentrations, (**B**) the maximum releasing concentration (mg/mL) of tilorone from its pure drug form and from (2, 5, and 10% *w*/*v*) PVA-CHO/CHIT-SH hydrogels, (**C**) in vitro release of tilorone dihydrochloride from its pure form and from different hydrogel (2, 5, and 10% *w*/*v*) concentrations in the presence of lysozymes, and (**D**) the maximum releasing concentration (mg/mL) of tilorone from its pure drug form and from (2, 5, and 10% *w*/*v*) PVA-CHO/CHIT-SH hydrogels in the presence of lysozymes.

**Table 1 pharmaceutics-14-02723-t001:** Interpretation of the release experiments using different kinetic models.

	Zero-Order Model		First-Order Model		Higuchi Model		Hixson–Crowell Model		Korsmeyer–Peppas Model		
Sample	r^2^	k (h^–1^)	r^2^	k (h^–1^)	r^2^	k (h^−1/2^)	r^2^	k (h^−1/3^)	n	r^2^	k (h^–n^)
Pure Tilorone	0.6602	12.496	0.8142	0.653	0.7841	35.375	0.5194	0.218	0.546	0.5745	41.029
2% PVA-CHIT	0.8273	11.266	0.9403	0.252	0.962	34.553	0.5293	0.373	0.649	0.3946	25.834
5% PVA-CHIT	0.9513	10.599	0.9946	0.181	0.9956	32.519	0.6979	0.428	0.906	0.6579	13.810
10% PVA-CHIT	0.9643	6.911	0.9901	0.096	0.9997	21.662	0.8091	0.307	0.909	0.702	9.687

**Table 2 pharmaceutics-14-02723-t002:** Interpretation of the release experiments using different kinetic models in the presence of lysozymes.

	Zero-Order Model		First-Order Model		Higuchi Model		Hixson–Crowell Model		Korsmeyer–Peppas Model		
Sample	r^2^	k (h^−1^)	r^2^	k (h^−1^)	r^2^	k (h^−1/2^)	r^2^	k (h^−1/3^)	n	r^2^	k (h^−n^)
Pure Tilorone	0.7444	8.3884	0.9717	0.442	0.8856	28.687	0.6367	0.1739	0.333	0.9153	57.042
2% PVA-CHIT	0.855	11.029	0.9664	0.243	0.982	35.049	0.5476	0.3867	0.7018	0.3991	22.563
5% PVA-CHIT	0.9108	10.55	0.9775	0.191	0.9794	33.389	0.6295	0.397	0.7272	0.4873	18.150
10% PVA-CHIT	0.8861	7.5437	0.934	0.116	0.972	24.429	0.7782	0.2791	0.739	0.5351	14.467

## Data Availability

Not applicable.

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
