# Peer review of "Self-Assembling Injectable Hydrogel for Controlled Drug Delivery of Antimuscular Atrophy Drug Tilorone"

_pharmaceutics, 2022, doi:10.3390/pharmaceutics14122723_

Round 1

Reviewer 1 Report

1. What's the polydispersity of PVA and chitosan?

2. The deacetylation of chitosan is around 75%, but in figure 1, it seems the deacetylation of chitosan is 100%.

4. Formatting issues. Line 112, 'C12H24N2O9' to 'C12H24N2O9'; line 139, 'Chitosan' to 'chitosan'. Please check all.

3. Line 51-52, several recent studies (doi.org/10.1021/acs.biomac.1c00086; doi.org/10.1016/j.cclet.2021.11.001) should be included to support such a claim.

5. Some of the figures are with framing line.

Reviewer 2 Report

Journal: Pharmaceutics

Title: Self-assembling injectable hydrogel for controlled drug delivery of anti-muscular atrophy tilorone

Comments: Abdelghafour et al studied the self-assembling capacity of hydrogel for the release of tilorone drug in a controlled manner in anti-muscular atrophy. The authors have prepared two components of injectable hydrogel and modified and characterized it. They also showed the in-vitro release of Tilorone from the hydrogel was controlled by the crosslinking density/concentration of the hydrogel. The manuscript is an interesting study.

Specific comments:

1. Abstract required more tangible

2. Introduction was vaguely written, and required to add more about issues in previous delivery methods.

3. Materials methods and results were written well with details.

4. Too little was discussed about their results and previous studies. 

Reviewer 3 Report

The proposed paper “Self-assembling injectable hydrogel for controlled drug delivery of anti-muscular atrophy tilorone” is devoted to the formation and study of PVA-aldehyde/ chitosan-amine+thiol cross-linked hydrogels as in situ gelling formulations for prolonged release of Tilorone. The paper contains many different results, however the characterization of used polymers chemical structure was not properly done. Also, the toxicity of hydrogels, which could contain highly toxic aldehyde groups, was not tested. Thus, any medical/pharmaceutical application of these hydrogels is very unclear and more theoretical, than practical. Thus, I would suggest resubmission of this MS to the journal with polymeric scope.

Comments:

11.  Introduction does not contain information and discussion on types of physical and covalent in situ forming hydrogels. The choice of covalent hydrogels is unclear.

22 .      Page 2, line 44: “…injectable materials have several benefits”. What are they? Authors didn’t list them out.

33.   Page 2, line 53: “A dynamic equilibrium between the Schiff base and the amine and aldehyde reactants is possible in an aqueous medium due to the more stable nature of the aromatic Schiff base bonds”. How more stable Schiff’s bases favor the equilibrium? They should shift the equilibrium to the right (formation of imine). The need for dynamic equilibrium in the case of in situ forming gels is also not evident, but not well-explained by the authors.

44.     Page 2, line 60: authors about chitosan “However, numerous drawbacks, such as dissolving in extremely acidic conditions, high cost, low surface area, and poor mechanical and thermal characteristics, limit its use.” So it is unclear why they have chosen chitosan for their experiments.

55.      The choice of non-degradable PVA adduct with 4-formyl benzoic acid is also not evident and poorly explained. There are many hydrophilic polymers like poly(N-vinylpirrolidone), poly(2-hydroxypropylmethacrylate), which are much more biocompatible than PVA.

66.      Page 3, line 110. PVA molecular weight is too high for non-biodegradable polymer for in vivo application. This polymer in the case of its systemic distribution could cause problems with kidney filtration.

77.       Page 3, Line 126. The reference for Steglich esterification was not provided.

88.       Page 3, line 140. When authors describe the synthesis of 3-mercaptopropionate chitosan it becomes clear, that 3-mercaptopropionic acid, chitosan, acetic acid and water-soluble carbodiimide (EDC) are present in the reaction medium. The authors propose the reaction between activated by EDC 3-mercaptopropionic acid carboxylic group and amino groups of chitosan. However, acetic acid could be also activated and participate in the reaction with amino-groups of chitosan. Moreover, thiol group of 3-mercaptopropionic acid is also good nucleophile and can react with activated carbonyl groups of both 3-mercaptopropionic acid (intermolecular reaction) and acetic acid. So, author consider only one electrophilic and one nucleophilic center in the reaction medium. However, there are two electrophilic centers and 2 nucleophiles. Authors did not provide information on the yield of the product. Also, they didn’t used NMR spectroscopy to prove the formation of the adduct. They used only FTIR spectroscopy (Page 7, line 305), which doesn’t prove anything in this case, because it could be just spectra of the mixture. Also, it is very hard to see anything in FTIR spectra, because formation of new bonds is not the case for the discussed mixture. It is obvious that carbonyl groups (non-deacylated moieties) were in the polymers before reaction and then after reaction. The magnification of the spectra does not allow to see the difference. I will repeat this one more time – it is impossible to prove and well describe the adducts without application of NMR spectroscopy. Thus, the chemical structure and composition of the polymers, which authors apply in their study is completely unclear. In my opinion, pharmaceutical studies should deal with chemically well characterized structures and samples, especially in the cases when it is quite easy to do. The degree of substitution was not provided by authors.

99.     Page 7, line 286, Figure 1 and Page 8, Line 330: “…PVA-CHO solution with CHIT-SH polymer resulted in the formation of the hydrogel at neutral pH due to the formation of the Schiff-base (imine bond)…” However, the reaction between aldehyde and thiol is also possible, as well as reaction between aldehyde and hydroxyl groups inside PVA-CHO macromolecules to form very stable acetals. CHIT-SH could react with CHIT-SH, both inter- and intra-molecularly to form -S-S- bonds. All these possibilities were not discussed by the authors. The uncertain chemistry of hydrogels formation leads to the bad understanding of further results.

110.   Page 8, line 343 “…viscosity was low due to imine bond deformation, …” What is imine bond deformation?

111.   Page 8, line 348: “The substitution of NH2 groups of initial CHIT by the thiol group led to a change in the polymer’s solubility since CHIT-SH was completely insoluble at pH 6 and totally soluble at pH 7 while partially soluble at pH 8 due to the isoelectric point and aggregate in alkaline conditions, respectively” It is hard to analyze this data without information on degree of substitution. Also, authors didn’t provide the explanation of the SH groups effect on solubility of chitosan. The effect is not evident.

112.   Page 9, line 356, Figure 3. The effect of acidic pH on reduction of -S-S- bonds should be discussed.

113.   Page 9, line 375. The method of membranes preparation for measurement of contact angles was not provided. The hydrophilicity/hydrophobicity of PVA is affected by preparation method. Sometimes the contact angle of PVA could be around 50-70 degrees. So, it is very unclear why it is so hydrophobic.

114.   Page 10, line 398, Figure 5. How the gel could be degradable if PVA is non-degradable? Only chitosan could be affected by lysozyme. This figure misleads the reader.

115.   Page 12, line 448. Self-healing was not discussed in introduction. The need for selfhealing of such materials, releasing the drug for less than 24 hours, is unclear.

116.   Page 14, line 527, Figure 10. The release of tilorone should be performed in the presence of enzyme. In this case the drug release and resorbtion (degradation) of the gel should correlate. Current results didn’t show such correlation. The release is governed only by diffusion (according to Higuchi, Page 15, line 544, Table 1). However, in vivo degradation should affect the release and the mechanism will be very different as well as release rate. The release experiments should better approximate the conditions of the actual application.

117.   Page 4, line 149 “Preparation of PVA-CHO / CHIT-SH hydrogels” also Page 6, line 247. The gel formation requires 1:1 water-ethanol mixture. How authors want to apply this system for in vivo in situ gel formation. It is not good to inject so much of ethanol to human body.

118.   Toxicity of aldehyde bearing materials was not tested. Aldehyde bearing moieties could be eliminated by materials and become free. This will make the material very toxic for surrounding tissues. Cytotoxicity is the must, and in vivo toxicity with rats is also needed to see the effect on the surrounding tissues.

Reviewer 4 Report

It was a great pleasure for me to review the manuscript numbered pharmaceutics-1971684. In this article, the Authors presented the self-assembling injectable hydrogel for controlled drug delivery of anti-muscular atrophy tilorone. In my opinion, this study is very interesting and will be of considerable interest to the readership of Pharmaceutics in the areas of drug delivery and hydrogel materials. The Authors carefully carried out all research and statistical analysis of the results. The article was written very clearly and correctly.

Therefore, I would like to recommend this article for publication in Pharmaceutics.

Round 2

Reviewer 3 Report

Authors have made really great work to improve their manuscript. I guess it could be accepted in its current form.